# Appearance and suppression of Turing patterns under a periodically forced feed

Brigitta Dúzs[1,6], Gábor Holló [2], Hiroyuki Kitahata [3], Elliott Ginder [4], Nobuhiko J. Suematsu [4], István Lagzi [2,5✉] & István Szalai[1✉]

Turing instability is a general and straightforward mechanism of pattern formation in reaction–diffusion systems, and its relevance has been demonstrated in different biological phenomena. Still, there are many open questions, especially on the robustness of the Turing mechanism. Robust patterns must survive some variation in the environmental conditions. Experiments on pattern formation using chemical systems have shown many reaction–diffusion patterns and serve as relatively simple test tools to study general aspects of these phenomena. Here, we present a study of sinusoidal variation of the input feed concentrations on chemical Turing patterns. Our experimental, numerical and theoretical analysis demonstrates that patterns may appear even at significant amplitude variation of the input feed concentrations. Furthermore, using time-dependent feeding opens a way to control pattern formation. The patterns settled at constant feed may disappear, or new patterns may appear from a homogeneous steady state due to the periodic forcing.

[1] Laboratory of Nonlinear Chemical Dynamics, Institute of Chemistry, Eötvös Loránd University, Pázmány Péter stny. 1/A, H-1117 Budapest, Hungary. [2] ELKH-BME Condensed Matter Research Group, Műegyetem rkp. 3, H-1111 Budapest, Hungary. [3] Graduate School of Science, Chiba University, Yayoi-cho 1-33, Inage-ku, Chiba 263-8522, Japan. [4] School of Interdisciplinary Mathematical Sciences, Graduate School of Advanced Mathematical Sciences, and Meiji Institute for Advanced Study of Mathematical Sciences (MIMS), Meiji University, 4-21-1Nakano Tokyo 164-8525, Japan. [5] Department of Physics, Institute of Physics, Budapest University of Technology and Economics, Műegyetem rkp. 3, H-1111 Budapest, Hungary. [6] Present address: University of Mainz, Duesbergweg 10-14, 55128 Mainz, Germany. ✉email: lagzi.istvan.laszlo@ttk.bme.hu; istvan.szalai@ttk.elte.hu

Pattern formation in biological systems is a complex phenomenon with many open questions, not only on the biochemical machinery but also on the general dynamics of such phenomena. The multiplex environmental ties that determine the development of organisms (changes in pH, mechanical stress, light intensity, etc.) deserve particular attention. How can such dynamic conditions lead to relatively static, stationary patterns (e.g., homeostasis, reproduction within a population, etc.)? As a start, in 1952 Alan Turing proposed a concept for generating stationary patterns (spots, stripes) from a uniform state by coupling chemical reactions of chemical species to their diffusion[1]. He also suggested that this approach can serve as a simple model of morphogenesis. The proposed two-variable model exhibits stationary pattern formation if the diffusivity of the two chemical species differs from each other, and there is initially a small inhomogeneity (perturbation) in the system. The experimental realization of such Turing patterns within a simple reaction network remained challenging and dormant for longer than had been expected. This was because it turned out that the ratio between the diffusivities should be at least greater than one order of magnitude for the generation of sustained stationary patterns in chemical systems.

The first experimental demonstration of sustained chemical Turing patterns was made using the chlorite–iodide–malonic acid (CIMA) reaction[2]. In the CIMA reaction, the iodide and chlorite ions drive the activatory and inhibitory processes. The critical issue of the pattern formation is the relatively slower diffusion of the activator than the inhibitor, which results from the reversible complex formation between the iodide, iodine, and starch (a very low mobility macromolecule). The reaction was performed in a two-sided-fed gel strip reactor (TSFR) with constant continuous feeding of the reagents and resulted in a row of equally spaced standing spots[2]. These pioneering results opened the way for the

experimental study of Turing pattern formations, which is still at the forefront of non-equilibrium self-organization[3–12]. Additionally, in the last three decades, it has been a widely accepted viewpoint in developmental biology that the pattern formations observed on the skin of animals (e.g., *Felidae*, fish) can be understood and described by the original Turing mechanism[13–19]. To prove this, several attempts have been made in vivo to understand and control fish skin pigmentation[20–27].

In most experimental, numerical, and analytical studies related to the generation of stationary Turing patterns, constant experimental conditions and parameters, including the inflow rate of reagents, have been used. However, in real biological systems, the parameters of the systems are not stationary. There is always a periodic change (forcing) on various parameters (e.g., temperature, illumination, the concentration of the chemical substances) having various time periods (diurnal, seasonal, yearly) dictated by the environment and biochemical cycle of the organisms. These parameter changes are much greater than the effect of fluctuations (e.g., thermal) against which the Turing systems are stable. Since the Turing mechanism plays a central role in pattern formation in biological systems[13–19], it is important to address whether periodic changes in one or more parameters can destabilize and/or alter the morphology of the formed pattern. Another interesting question is whether the periodic forcing of parameters can facilitate the generation of a stable pattern.

Some aspects of global periodic forcings and Turing pattern formation dynamics have been demonstrated recently by the periodic illumination of the light-sensitive chlorine-dioxide–iodine–malonic acid (CDIMA) reaction, which is the core part of the CIMA reaction[28–34]. The experiments demonstrated that periodic illumination might suppress the patterns, especially at a frequency equal to the frequency of autonomous oscillations in a well-stirred reactor[28]. The detailed exploration of the spatially resonant forcing and the non-resonant case revealed entrained and oscillating patterns[33]. It was also demonstrated, both experimentally and theoretically, that using periodic masks to control the spatial period of the illumination can produce a superlattice Turing pattern[34]. All these observations have been made under global forcing, where the whole spatially distributed system was forced by illumination of the part of the applied reactor where patterns could develop. Recently, time-dependent temperature control of the boundary has been suggested theoretically as a way to control the formation of Turing patterns[35].

Here we will show the effect of a different external parameter: the sinusoidal modulation of the inflow rate of one of the reagents in a Turing system, which generates a periodic change in the feed concentration. This periodic forcing can destabilize the pattern generated by a constant inflow rate and generate stationary patterns from the no-pattern regime, depending on the average inflow rate and its amplitude (Fig. 1). To support our experimental findings, we performed numerical simulations of the corresponding reaction−diffusion (RD) equations with a periodic forcing and a semi-analytic linear stability analysis of the system.

## Results and discussion

**Experimental findings**. The experiments were performed in a two-sided-fed open gel reactor, where the separated chemicals diffuse into the hydrogel from two tanks (A, B) placed on the opposite sides of the gel (Fig. 2a)[36]. This configuration creates counter-concentration gradients of the chemicals, and the reactions only take place in a tiny region inside the gel, parallel to the feed surfaces. The experiments were started from a steady state by applying constant inflow rates of the reagents in both tanks, thus creating constant (feed-) concentrations on both sides of the gel. The control parameter in the experiments was the feed

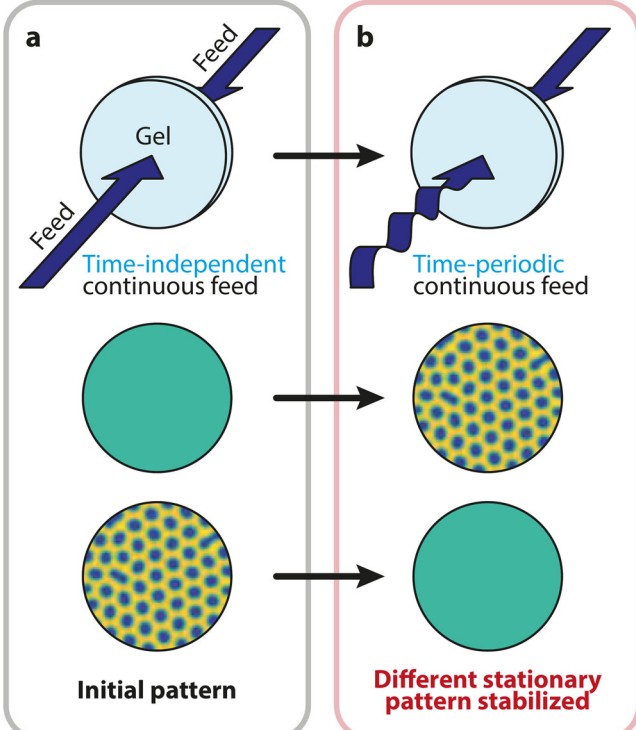

**Fig. 1 Concept of controlling Turing patterns by periodic forcing.** Different initial patterns can be stabilized at constant feed concentrations (**a**). During the continuous periodic forcing of the input feed concentration of one reactant drastically different pattern forms in the system (**b**).

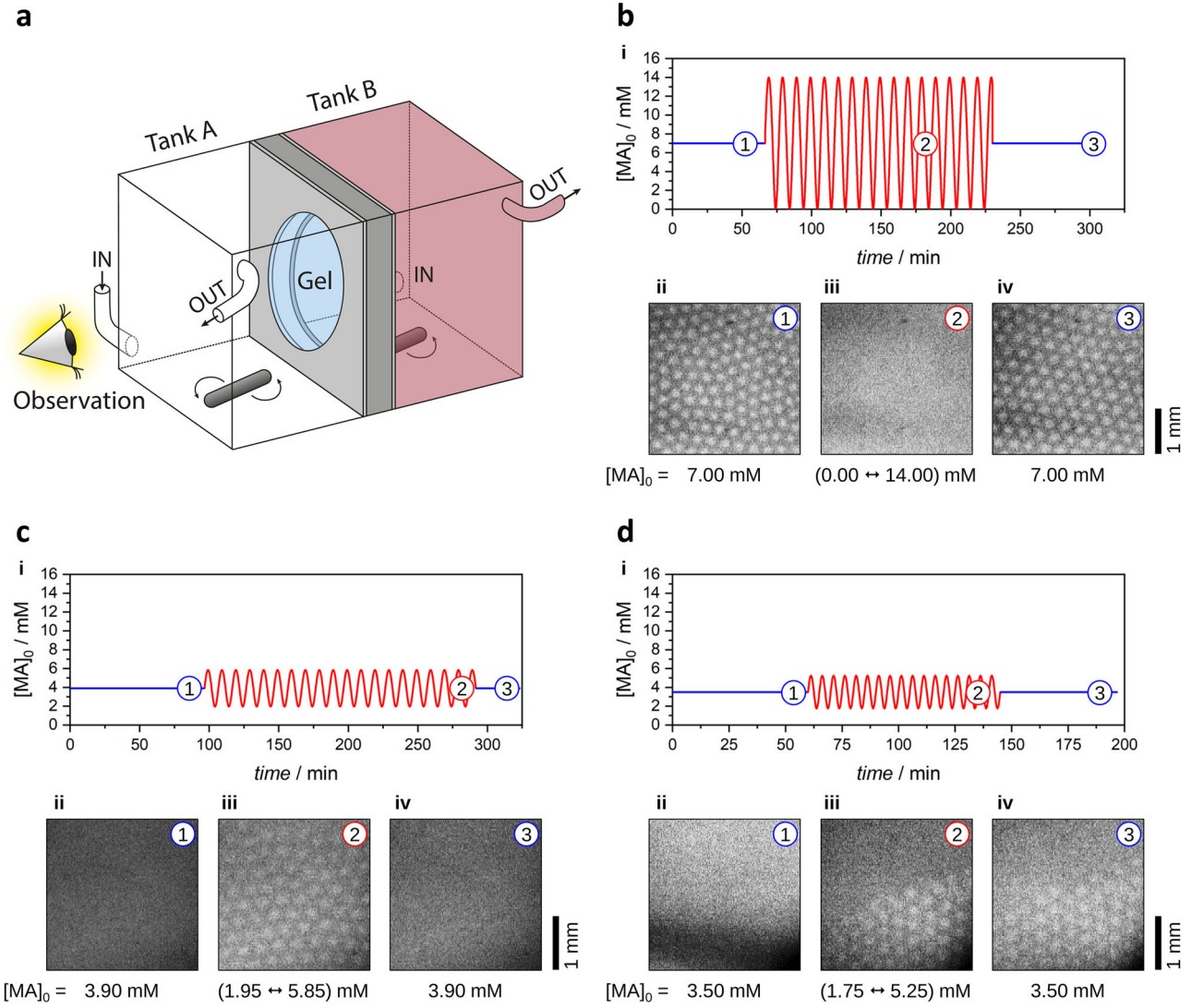

**Fig. 2 Periodic forcing of the feed concentration of MA in experiments.** The sketch of the experimental system (**a**). Disappearance of stationary Turing spots induced by the periodic forcing of the feed concentration of the MA. The initial $[MA]_{0,basis} = 7.00$ mM was modulated with ±100% amplitude and a time period of 10 min (**b**). Generation of stationary Turing spots induced by the periodic forcing of the feed concentration of the MA. The initial $[MA]_{0,basis} = 3.90$ mM was modulated with ±50% amplitude and a time period of 10 min (**c**). The appearance and locking of stationary Turing spots induced by the periodic forcing of the feed concentration of the MA. The initial $[MA]_{0,basis} = 3.50$ mM was modulated with ±50% amplitude and a time period of 5 min (**d**). In figures **b–d**: (i) change of the concentration of the MA in the tank ($[MA]_0$) during the whole experiment. (ii)–(iv) Snapshots of the stabilized behavior with constant $[MA]_0$ before the forcing (1), periodic $[MA]_0$ during the forcing (2), and constant $[MA]_0$ after the forcing (3).

concentration of the malonic acid (MA) in tank A ($[MA]_0$), which could be tuned by changing the inflow rate of this reagent.

First, to explore the general dynamical behavior of the chemical system, we carried out experiments using constant feed concentrations of the reagents. In this set of experiments, only the inflow rate of the MA was changed (but it was fixed through each experiment). We observed the following states by increasing the feed concentration of MA (for a detailed diagram, see Supplementary Fig. S1). When the $[MA]_0$ was below 4 mM, no pattern formation was observed. At 4 mM, bistability was found between the homogeneous no-pattern state and Turing spots. Above this value, stationary spot patterns with a hexagonal symmetry appeared, as shown in Supplementary Fig. S1b. As $[MA]_0$ was further increased, the formation of stationary stripes and spatiotemporal waves could be observed. Above $[MA]_0 = 10$ mM, the waves disappeared, and the gel showed no pattern formation. The bistability between the homogeneous state and Turing spots is quite narrow and sensitive to the small

differences in the actual gel thickness that naturally varies a bit (~0.1 to 0.2 mm) from experiment to experiment. The boundaries between the regimes of spots and mixed spots-stripes were also sensitive to minor disturbances, inhomogeneities in the gel. However, we detected real bistability between the Turing structures (spots, mixed spots-stripes) and the waves, i.e., the overlap of the regimes of these dynamical behaviors between $[MA]_0 = 5.5$ and 7.5 mM. Our observations correspond to the results reported earlier by De Kepper and his coworkers[37].

A typical experiment applying the periodic forcing had three stages. First, we created constant feed concentrations for all reagents, including MA. Secondly, once the corresponding state (which could be homogeneous or patterned) was stabilized (~1 h), the periodic forcing was applied. This was implemented in a sinusoidal modulation of the inflow rate of the MA in one tank. Lastly, after the emergence of the state corresponding to the periodic forcing (~1 h), the periodic forcing was switched off, and a new state was developed, which, in several circumstances,

differed from the initial one (before the periodic forcing), even though the same inflow rate was applied for the MA. Here we present nontrivial cases, where the periodic forcing induces quasi-stabilized behaviors, which have a time-independent characteristic wavelength.

During our investigation, we explored several characteristic scenarios in the experiments. In the first case (Fig. 2b, Supplementary Fig. S2, and Supplementary Movie 1), the sinusoidal forcing on $[MA]_0$ was switched on after the hexagonal spot pattern had already been settled at a constant 7 mM of $[MA]_0$. A large-amplitude modulation on $[MA]_0$, where the concentration of MA in the tank varied between 0 and 14 mM, entirely and quickly suppresses the Turing pattern. After eliminating the forcing, the system returns quickly to the original patterned state. That is a clear sign of the robust nature of Turing patterns in this RD system. In the second case (Fig. 2c, Supplementary Fig. S3, and Supplementary Movie 2), the sinusoidal forcing on $[MA]_0$ was switched on when the RD system in the gel showed a homogeneous no-pattern state at 3.9 mM of $[MA]_0$. This value is slightly below the critical value where patterns appear spontaneously in the gel due to the Turing bifurcation. A moderate amplitude sinusoidal forcing, where $[MA]_0$ varied between 1.95 and 5.85 mM, induced the formation of a hexagonal spot pattern. The periodic forcing maintained the pattern, and the pattern disappeared when the constant inflow rate of MA was used again. As the transition from the homogeneous state to the spot patterned state is generically subcritical, bistability between the homogeneous and the patterned state is expected, but it is difficult to observe it experimentally[37]. Our findings show that the sinusoidal forcing can facilitate and stabilize the pattern formation. The third scenario was observed at lower $[MA]_0$ (3.5 mM), where the RD system in the gel showed no pattern. The hexagonal spot pattern formed after the modulation was applied, and the $[MA]_0$ varied between 1.75 and 5.25 mM. Contrary to the previous case, the patterned state remained stable even after switching off the periodic forcing (Fig. 2d, Supplementary Fig. S4, and Supplementary Movie 3). In this case, the color contrast of the forming pattern was smaller than in the previous cases, thus pictures were recorded with enhanced contrast. The appearance of the black zone at the bottom of Fig. 2d and Supplementary Movie 3 is the result of the prompt mixing of the inlet solutions in the tank in front of the gel. This is naturally visible due to the applied camera settings; however, the mixing was immediate and did not cause any unwanted gradients in the gel during this experiment.

The amplitude and period of the forcing of the above experiments were selected using the following considerations: (i) the forcing frequency should fit the inherent frequency of the unperturbed spatiotemporal oscillations; (ii) the forcing frequency should fit the response time of the system to variations in the boundary conditions; (iii) the tank damps the forcing, which is significant when the forcing period is shorter than the residence of the stirred tank reactor[38]. Swinney and coworkers found that the typical period of oscillations in the CIMA reaction near a Hopf bifurcation is between 9 and 60 s[39]. According to our experiences and the published data in the literature[40], the patterns in the CIMA reactions respond relatively rapidly, within a few minutes, to variations in the boundary conditions. The period was chosen to be higher than the residence time of tank A (that is 2–3 min), otherwise, the tank itself (and also the gel) would have been damped the forcing. Also, it should have been shorter than 40–60 min that is typically enough for the pre-stabilization of the patterns, because in such a case the forcing would have resulted in two alternating behaviors instead of one quasi-stabilized one. Thus, we have arbitrarily chosen $T = 5$ and 10 min in the discussed regime. Based on the numerical simulations (see the

next chapter), we made experiments with forcing amplitudes of 50% and 100% of the initial $[MA]_0$. The simulations suggested that the regime of the interesting forcing-induced phenomena is not limited to a very narrow range of amplitude, so in the experiments, we stuck to the values of 50% and 100% of the initial $[MA]_0$ and made systematic screenings varying the initial $[MA]_0$.

**Numerical simulations**. In a TSFR, Turing patterns form in a layer perpendicular to gradients imposed by the boundary conditions. The size of our reactor in the perpendicular dimensions is significantly larger than in the direction of the gradient. The effect of the gradient in forming Turing patterns in the CIMA reaction has been studied experimentally[37,41] and theoretically[40,42]. These studies pointed out that in the inherently three-dimensional system, the strong anisotropy forms quasi-two-dimensional patterns. Therefore, reduced two-dimensional simulations show reasonable qualitative agreement with the experimental observations[43]. This does not mean that a three-dimensional effect cannot occur under some conditions, as numerical simulations made in parameter gradients show the possibility of forming different patterns at different locations along the gradients[44,45]. We have not observed clear signs of three-dimensional effects in our experiments. Therefore, we choose to use reduced two-dimensional modeling, where the feeding from the boundary is described by parameters that appear in the partial differential equation (PDE, Eq. (1)).

The other issue that must be considered is the level of description of the actual chemistry. Quantitative modeling of the CIMA reaction could be made by the nine-variable model of Lengyel and coworkers[46] or at least by a five-variable version[43]. A chemically reasonable simplification of these detailed models to a two-variable one was proposed by Lengyel and Epstein[43]. This model shows the structure of a general activator-inhibitor system and allows the numerical and analytical study of pattern formation.

To support our experimental findings, we performed numerical simulations applying the periodic forcing to the parameter $a$ (which is the feed concentration and connected to the experimental $[MA]_0$) in the Lengyel−Epstein model[4]. The phenomenon can be described by the following extended set of PDEs for the non-dimensional concentration of iodide ($u$) and chlorite ($v$):

$$\frac{\partial u}{\partial t} = a + A \sin\frac{2\pi}{T} t - u - \frac{4uv}{1+u^2} + \nabla^2 u \qquad (1)$$

$$\frac{\partial v}{\partial t} = \sigma\left( b\left( u - \frac{uv}{1+u^2} \right) + c\nabla^2 v \right) \qquad (2)$$

where $A$ and $T$ are the amplitude and the time period of the periodic forcing of the feed concentration, respectively, and we impose periodic boundary conditions. We note that when $A = 0$, the model equation corresponds to the original system described by Lengyel and Epstein. In the numerical simulations, first, we explored the effect of constant $a$ on the pattern structure (Supplementary Fig. S5). We used 1.9 for the value of parameter $b$ obtained from the estimation using experimental data (for the details of the estimation, see the section "Methods"). We found that starting from a homogeneous initial state, Turing pattern formation occurs between $a = 23$ and $a = 27$ from spots to stripes, while below $a = 23$ and above $a = 27$, stationary homogeneous patterns and homogeneous oscillations exist, respectively. This finding is in good agreement with the results of our analytical investigation and experiments (in experiments, the Turing pattern exits between $a \sim 23$ and $a \sim 54$). However, once the patterns have been settled, they persist for a wide range of parameters in the numerical model. Accordingly, we observed bistability between the homogeneous state and Turing spots and

also between Turing spots and stripes (Supplementary Fig. S6), but the experimental observation of the latter one is unlikely[47].

The extended numerical model developed here could capture the experimentally observed pattern transition stages once the periodic forcing was applied. The stationary spotted pattern generated at constant feed concentration could be erased by applying a high amplitude periodic forcing (Fig. 3a and Supplementary Movie 4). Once the forcing was switched off, the original Turing pattern emerged. When the periodic forcing started from the no-pattern stage, moderate small amplitude modulation of the feed concentration could induce the Turing pattern. After switching off the forcing, the produced pattern was locked (Fig. 3b and Supplementary Movie 5). It should be noted that once the forcing was on, the generated pattern exhibited pulsation, so this pattern is not stationary in a strict sense; however, the corresponding wavenumber (wavelengths) did not change during the modulation of the feed concentration. Phase diagrams showing the period and amplitude range of forcing-induced behaviors are presented for both above cases (Supplementary Fig. S7).

In simulations, we could not produce the no-pattern—pattern—no-pattern transition observed in experiments. One plausible explanation of this finding can be the limited power of the kinetic equations used in the Lengyel−Epstein model. This was introduced to reproduce the Turing pattern formation in the case of a constant set of parameters, including the feed concentration of the reagents. The chemical mechanism of the Turing pattern formation can be much more complicated, which is not perfectly captured in the Lengyel−Epstein model when a periodic forcing is introduced. However, the numerical simulations revealed other types of pattern transitions, namely the transition from the stationary stripes to no-pattern regime and going back to stripes (Supplementary Fig. S8a and Supplementary Movie 6) and the transition from striped-spotted pattern to spotted one (Supplementary Fig. S8b and Supplementary Movie 7).

**Linear stability analysis**. To gain more understanding of the experimentally and numerically observed dynamics we performed linear stability analysis, which provides general information about the stability of the different states of the system against small perturbations. Here we will discuss the results of a semi-analytic approach to understanding the linear stability of uniform oscillatory states. Although a fully analytic treatment is desirable, we remark that the model seems to evade the application of such approaches. In particular, we attempted to apply the sufficient condition for instability given in ref. [48] to our model equation. However, it was observed that certain wavenumbers $k$ fail to satisfy the corresponding inequality. This fact has led us to employ a semi-analytic approach.

In our investigation, we perform the linear stability analysis on the uniform oscillatory solution for the system in Eqs. (1) and (2). Here, we fix the parameters as $a = 22$, $\sigma = 8$, $b = 1.9$, and $c = 1.5$. It should be noted that the Hopf bifurcation point and the Turing bifurcation point with respect to $a$ (i.e., at fixed $\sigma$, $b$, and $c$) are approximately located at $a = 26.88$ and $a = 22.07$, respectively. First, we numerically obtain the uniform periodic solution (Fig. 4a and b where Fig. 4a shows one period of $u_p(t)$ and $v_p(t)$). In Fig. 4, we set $u(0) = a/5$, $v(0) = 1 + a^2/25$, $A = 6$, and $T = 1$. Then $u_p(t)$ and $v_p(t)$ are obtained as the large-time behavior of Eqs. (1) and (2), using a step size $\triangle t = T/N$, where $N$ takes the value 10,000. Using $u_p(t)$ and $v_p(t)$, we then construct the matrix $\mathbf{P}(k)$, which satisfies

$$^{\text{t}}(\delta u^{(k)}(T), \delta v^{(k)}(T)) = \mathbf{P}(k)^{\text{t}}(\delta u^{(k)}(0), \delta v^{(k)}(0)) \qquad (3)$$

where $^{\text{t}}w$ means the transpose of the vector $w$. $\mathbf{P}(k)$ is a linearized Poincare map for a period $T$, which shows the time evolution of

the perturbation with the wavenumber $k$ imposed to a uniform oscillatory solution. We determine the absolute values of its eigenvalues $|\lambda_1(k)|$ and $|\lambda_2(k)|$. Figure 4c displays the magnitude of the eigenvalues with respect to the wavenumber $k$. The numerical results show that the uniform periodic solution is unstable over a range of wavenumbers. We also investigated the behavior of the dominant eigenvalue $|\lambda(k)|_{\max}$ as the amplitude of the external force $A$ is varied. The results are shown in Fig. 4d, where we can see that the uniform periodic solution destabilizes at around $k = 1.2$ for $A \gtrsim 2$. By analyzing the numerical behavior over a wider range of parameter values, we find that the stability of the uniform oscillatory state behaves similarly to what is often observed in resonance-induced instabilities. In particular, for certain amplitudes $A$, the length of the period $T$ can be varied to traverse from regions of stability to those of instability (the corresponding phase diagram is shown in Fig. 5).

## Conclusion

In summary, we have demonstrated that several pattern-formation phenomena can arise and persist when parameters and environmental conditions change in a well-defined periodic manner. This time dependence of the feed concentration can be used to control the state of the reaction-diffusion system. Continuously fed open gel reactors are perfect tools to study the dynamics of an RD system. In this setup, complex structures can be sustained for arbitrarily long, such as perfectly ordered hexagonal Turing spots, equidistant stripes, and combinations. However, such long-lasting constant boundary conditions are rare in biological systems, and cyclic environmental conditions and quasi-periodic reagent supply are more relevant for biomimetic applications. Understanding how the out-of-equilibrium systems react to time-periodic boundary conditions is a central question that has been investigated only sporadically[35,38]. Two extremities would be to follow the governing periodic signal perfectly or to balance the outer periodic forcing and remain undisturbed (keeping homeostasis). In this study, we have addressed a third, highly nontrivial adaptation scenario, where the stationary behavior of the dynamic system (i.e., time-independent spatiotemporal pattern) switches to a qualitatively different one due to a periodic modulation of a feed concentration. Mechanisms leading to these highly programmed behaviors are essential for designing new types of active and adaptive materials systems.

We found that the disappearance of the stationary pattern can be facilitated in the spot regime with a higher amplitude of periodic forcing. On the other hand, we observed a transition, starting from a feed concentration where no pattern was observed, with a moderate amplitude of forcing, to Turing patterns. The pattern shows pulsation under periodic forcing, but this does not affect the planform and the wavelength. We performed linear stability analysis on the uniform periodic solution and discussed the onset of the pattern formation under periodic forcing. In standard stability analysis, we consider the stationary uniform state, and growth rates for the perturbation with a certain wavenumber are then calculated. In contrast, here the growth rates for the perturbation of certain wavenumbers are calculated with respect to the uniform oscillatory state (i.e., it is no longer stationary). Therefore, to evaluate growth rates, we must define a map for the time evolution over its period, and then determine the time evolution of the perturbation as a map. The advantage of the present method is that we can discuss the stability of the system without directly calculating the PDE. This allows us to vary the parameters and investigate the mathematical structure of the instability. By varying the amplitude and frequency, we see that the instability of the oscillatory uniform states under periodic forcing can easily occur within certain frequency ranges. The dependence is nonlinear, and this may reflect the intrinsic mathematical structure of the system. However, we

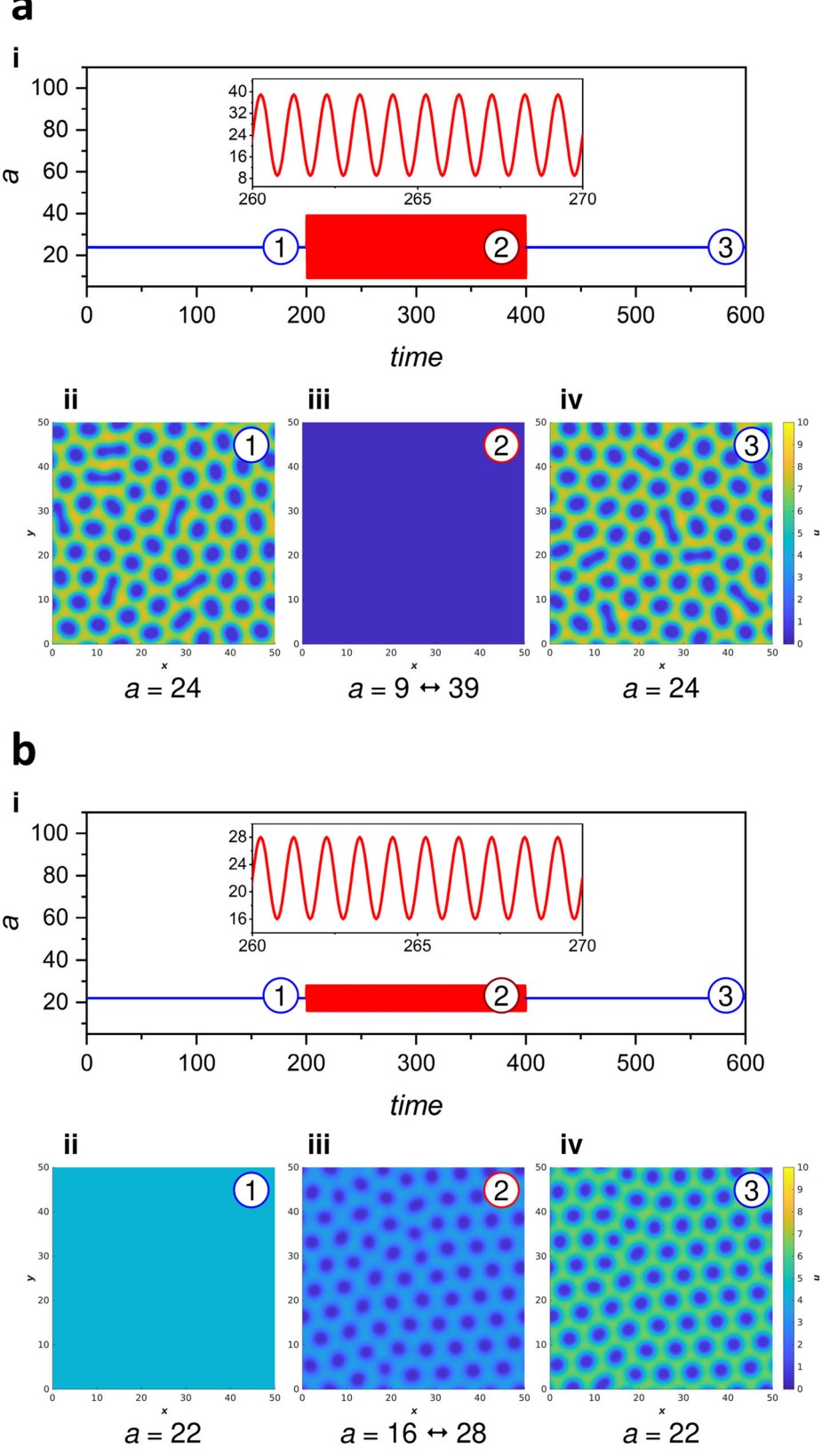

**Fig. 3 Periodic forcing of the feed concentration a in numerical simulations.** Disappearance of stationary Turing spots induced by the periodic forcing of the feed concentration $a$. The initial $a = 24$ was modulated with an amplitude of 15 and a time period of 1 (**a**). The appearance and locking of stationary Turing spots induced by the periodic forcing of the feed concentration $a$. The initial $a = 22$ was modulated with an amplitude of 6 and a time period of 1 (**b**). In figures **a** and **b**: (i) change of the feed concentration $a$ during the whole simulation. (ii)–(iv) Snapshots of the pattern generated with constant $a$ before the forcing (1), periodic forcing of $a$ (2), and constant $a$ after the forcing (3). The following parameter set was used: $\sigma = 8$, $b = 1.9$, $c = 1.5$, $\triangle x = \triangle y = 1.25 \times 10^{-1}$ (grid spacing), and $\triangle t = 3.124 \times 10^{-4}$ (time step). The length of the simulation domain and the simulation time were $50 \times 50$ and 600, respectively. All parameters are dimensionless.

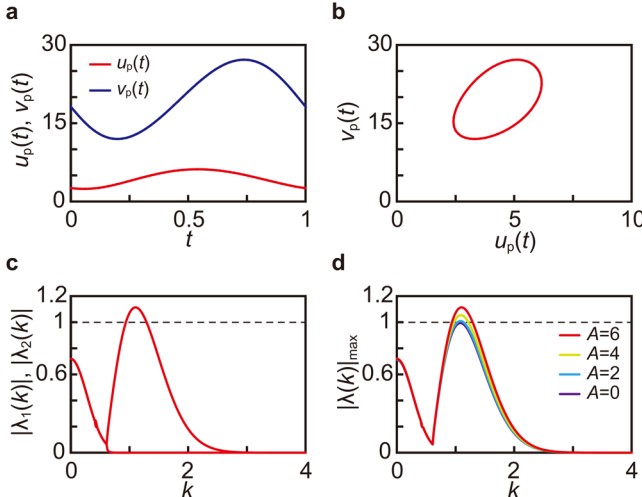

**Fig. 4 Linear stability analysis of the Turing system with a periodic forcing of the feed concentration.** Plots of $u_p(t)$ (red) and $v_p(t)$ (blue) for $A = 6$ (**a**). Plot of the or bit of $u_p(t)$ and $v_p(t)$ in the phase space ($u$–$v$ plane) for $A = 6$ (**b**). Plots of $|\lambda_1(k)|$ and $|\lambda_2(k)|$ as a function of $k$ for $A = 6$ (**c**). Plots of $|\lambda(k)|_{max}$ as a function of $k$ for various $A$ (**d**). The other parameters were set as $a = 22$, $\sigma = 8$, $b = 1.9$, $c = 1.5$, $T = 1$, and $N = 10,000$.

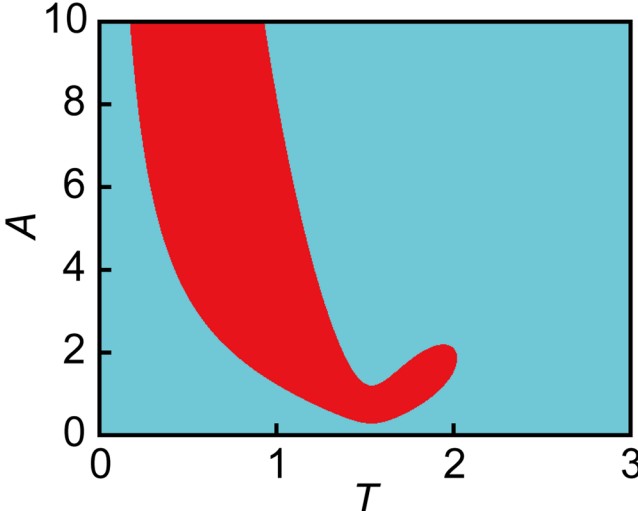

**Fig. 5 Phase diagram showing the stability of the uniform oscillatory state in the $T$-$A$ plane obtained by the linear stability analysis.** The cyan and red regions represent the stable and unstable regions, respectively. The parameters are set as $a = 22$, $\sigma = 8$, $b = 1.9$, $c = 1.5$, and $N = 10,000$.

cannot discuss the selection mechanism of the pattern, i.e., spot or stripe pattern. We also leave the stability analysis on the disappearance of the pattern by the periodic forcing. For such analyses, we have to develop new techniques, which presents an interesting area for furthering this study.

Understanding the robustness and the control of pattern formation is a critical issue in the biological context[49]. Our work opens up a variety of future directions to study this problem both experimentally and theoretically.

## Methods

**Experimental setup.** We used the classical disc-shaped two-side-fed gel reactor (TSFR) presented in previous works (Fig. 2a)[35,36]. The gel disc (the reaction medium) with a thickness of $w = 2.5$ mm and a diameter of $d = 20$ mm was made of 4 m/V% agarose (Sigma-Aldrich A0169). It was fixed between two separated,

continuously stirred, and fed liquid tanks, each with a volume of 26.0 mL. The gel was fed through its opposite surfaces via diffusion from the tanks, which provided fixed or periodically programmed boundary concentrations. The whole setup was immersed in a thermostatic water bath, and the experiments were made at $6.0 \pm 0.1\,°C$. The tanks were fed from solutions (1–4), which were stored at $6.0\,°C$ in separate reservoirs but entered premixed into the tanks. Tank A was fed from solutions (1) and (2), and its content was transparent. Solution (1): $NaClO_2$ (Sigma-Aldrich, puriss. p.a., 80%), polyvinyl alcohol (PVA, Sigma-Aldrich, $M_w = 9000$–$10,000$ g/mol, 80% hydrolyzed); solution (2): $KIO_3$ (Sigma-Aldrich, ≥98%), NaOH (VWR, ≥99%), PVA. Tank B was fed from solutions (3) and (4); its content was reddish due to the small amount of PVA–$I_3^-$ complex. Solution (3): KI (VWR, ≥99%), $H_2SO_4$ (VWR, 2.5 M), PVA; solution (4): KI, $H_2SO_4$, PVA, malonic acid (MA, Sigma-Aldrich, ≥99%). Solutions (1–3) were fed with constant-speed HPLC pumps (Pharmacia P500), and solution (4) was fed with a programmable peristaltic pump (Gilson Minipuls 3). The fixed concentrations in the respective tanks were $[NaClO_2]_0 = 10$ mM, $[KIO_3]_0 = 2$ mM, $[NaOH]_0 = 12$ mM, $[KI]_0 = 2$ mM, $[H_2SO_4]_0 = 10$ mM, $[MA]_0$ was varied between 0–14 mM, $[PVA]_0 = 1.5$ g/L. Depending on the experiment, we fed solution (4) with a constant inflow rate ($q_{1,basis}$, no forcing) or with the following sinusoidal inflow function ($q_1(t)$, periodic forcing):

$$q_1 = q_0 + A \sin\left(\frac{2\pi}{T}t\right) \tag{4}$$

where $q_1(t)$ and $q_0$ are the resulting and initial inflow rates during the sinusoidal perturbation, $A$ is the amplitude of the forcing, $T$ is the period. The constant or sinusoidal inflow rate of solution (4) resulted in a constant ($[MA]_{0,basis}$) or periodically changing ($[MA]_0(t)$) concentration of MA in tank B, respectively. The periodically modulated $[MA]_0(t)$ was calculated with the following differential equation:

$$\frac{d[MA]_0}{dt} = [MA]_s \frac{q_1(t)}{V} - [MA]\left(\frac{q_1(t)}{V} + \frac{q_2}{V}\right) \tag{5}$$

where $[MA]_s$ is the MA concentration of solution (4), $q_2$ is the flow rate of the constant pump feeding solution (3) to tank B, and $V$ is the volume of tank B. In our experiments, the volume of each tank was $V = 26$ mL, $q_2 = 499$ mL/h, and the sinusoidal forcing was made with $q_0$ varied in the range of 90–170 mL/h. In case of lower or higher desired [MA] concentrations, we used $[MA]_s = 20$ or 45 mM stock solutions, respectively. With this, we could keep the $q_0$ in the abovementioned optimal range. The $A$ of the modulation varied between 50–170 mL/h. The MA concentration in tank B right at the beginning of the sinusoidal perturbation was $[MA]_0(t = 0) = [MA]_s \frac{q_1(t=0)}{q_1(t=0)+q_2}$. The $q_{1,basis}$ and the $q_0$ were not the same: the $q_0$ and $A$ of the modulation were chosen to fulfill Eq. (6), i.e., to provide symmetric concentration modulation:

$$[MA]_{0,max} - [MA]_{0,basis} = [MA]_{0,basis} - [MA]_{0,min} \tag{6}$$

where $[MA]_{0,min}$ and $[MA]_{0,max}$ are the lowest and highest concentrations of MA in tank B during the periodic forcing, respectively. With this, we provided the same change to the positive and negative direction in $[MA]_0$ during the modulation compared to the nonperturbed $[MA]_{0,basis}$ value. This term is important since due to the finite volume of the tank, the temporal change of $[MA]_0$ was not perfectly sinusoidal; however, the distortion was negligible in the applied conditions. The average of the modulated $[MA]_0$ slightly increased compared to the constant inflow rate case, but it varied between 2.6% and 9.3% depending on $q_0$ and $A$ (Supplementary Fig. S9). It should be noted that this increase in the average of the $[MA]_0$ due to the sinusoidal modulation would not have created a new pattern state. The residence time of the other reagents in tank B also changed slightly, but it was less than 10%. In a TSFR, having nonreactive solutions separated in tanks A and B, such a distortion had no significant effect on the dynamics, and the gel was also strong enough to tolerate the slight change of pressure during the periodic modulation. During the periodic forcing, the average residence time in the tanks was $\tau_{basis} = 2$–3 min, and the period of forcing was $T = 5$ or 10 min. For an effective forcing, $T$ should be higher than $\tau_{basis}$; otherwise, the tank damps the modulation. The setup was enlightened with a white LED backlight (Advanced Illumination) through a 600 nm band-pass filter from the direction of tank A, and the pictures were recorded with a digital camera (Imaging Source DMK 33UX250) from the direction of tank A (Fig. 2a). The image processing was made by the ImageJ software.

**Estimation of the parameters of the numerical model using experimental data.** In the CIMA reaction, the important intermediates are $I^-$ (iodide ion) and $ClO_2^-$ (chlorite ion). To understand the mechanism of pattern formation, a mathematical model was developed by Lengyel and Epstein (Lengyel−Epstein model) with the following essential chemical reactions[4]:

$$MA + I_2 \rightarrow IMA + I^- + H^+ \tag{7}$$

$$ClO_2 \cdot + I^- \rightarrow ClO_2^- + 0.5I_2 \tag{8}$$

$$ClO_2^- + 4I^- + 4H^+ \rightarrow Cl^- + 2I_2 + 2H_2O \tag{9}$$

Here IMA is iodomalonic acid. This chemical reaction set can be simplified to the following skeleton mechanism:

$$A \xrightarrow{k_1} X \tag{10}$$

$$X \xrightarrow{k_2} Y \tag{11}$$

$$4X + Y \xrightarrow{k_3} P \tag{12}$$

where A, X, Y, and P are MA, $I^-$, $ClO_2^-$, and products, respectively. $k_1$, $k_2$, and $k_3$ are the reaction rate constants. The reaction rates of reactions (7)–(9) were measured and can be expressed as

$$v_1 = \frac{k_{1a}[I_2]_0}{k_{1b} + [I_2]_0}[MA]_0 \tag{13}$$

$$v_2 = k_2[ClO_2\cdot]_0[X] \tag{14}$$

$$v_3 = k_3[I_2]_0 \frac{[X][Y]}{\alpha + [X]^2} \tag{15}$$

where $[\cdot]_0$ denotes the concentration of the chemical species in the tank reactor (feed concentration), $k_{1a}$, $k_{1b}$, $k_2$, and $k_3$ are the reaction rate constants of reactions (7)–(9), and $\alpha$ is a reaction kinetic parameter. Coupling the reaction mechanism and its kinetics with the diffusion, we can obtain the following set of partial differential equations describing the evolution of concentrations of the intermediate species, X and Y:

$$(1 + K)\frac{\partial[X]}{\partial t} = k_1[MA]_0 - k_2[ClO_2\cdot]_0[X] + 4k_3[I_2]_0\frac{[X][Y]}{\alpha + [X]^2} + D_X\nabla^2[X] \tag{16}$$

$$\frac{\partial[Y]}{\partial t} = k_2[ClO_2\cdot]_0[X] - k_3[I_2]_0\frac{[X][Y]}{\alpha + [X]^2} + D_Y\nabla^2[Y] \tag{17}$$

Here $D_X$ and $D_Y$ are the diffusion coefficients of the intermediate species, $k_1 = \frac{k_{1a}[I_2]_0}{k_{1b}+[I_2]_0}$, and K is the equilibrium constant of the formation of the triiodide complex ($S + I^- + I_2 \rightarrow SI_3^-$), where S is typically starch[4]. The dimensionless RD model can be derived as

$$\frac{\partial u}{\partial t} = a - u - 4\frac{uv}{1 + u^2} + \nabla^2 u \tag{18}$$

$$\frac{\partial v}{\partial t} = \sigma\left(b\left(u - \frac{uv}{1 + u^2}\right) + c\nabla^2 v\right) \tag{19}$$

with the following dimensionless parameters $a = \frac{k_1[MA]_0}{k_2[ClO_2\cdot]_0\sqrt{\alpha}}$, $b = \frac{k_3[I_2]_0}{k_2[ClO_2\cdot]_0\sqrt{\alpha}}$, and $c = \frac{D_Y}{D_X}$. In the above, u and v denote the concentrations of the activator (iodide) and inhibitor (chlorite), respectively. Parameters a and b relate to the feed concentrations, namely, a higher value of a represents an increased supply rate of malonic acid relative to the supply of chlorine dioxide. Similarly, increasing b corresponds to a higher supply rate of iodine, and $\sigma = 1 + K$, which is a rescaling parameter that depends on the concentration of the starch. The last terms in Eqs. (18) and (19) describe the diffusion of the activator and the inhibitor. We used the following parameter values: $\alpha = 10^{-14}M^2$, $k_{1a} = 7.5\times10^{-3}s^{-1}$, $k_{1b} = 5\times10^{-5}$ M, $k_2 = 6\times10^3M^{-1}s^{-1}$, $k_3 = 2.65\times10^{-3}s^{-1}$, $\sigma = 8$, and $c = 1.5$[43].

To estimate the parameters of a and b in the Lengyel–Epstein model, the concentrations of MA, $ClO_2\cdot$, and $I_2$ in the tank reactor are needed. Since $ClO_2\cdot$ and $I_2$ are intermediate species, their concentrations can be estimated in the following manner. $ClO_2\cdot$ is produced by the reaction of $ClO_2^-$ and $I_2$. It has been reported that the chlorite–iodine reaction rapidly reaches the equilibrium state, usually within a couple of seconds[43]. The kinetic investigation of the chlorite–iodine reaction revealed that the amount of product ($ClO_2\cdot$) is roughly proposed to the amounts of reactants ($ClO_2^-$ and $I_2$). If there is an excess of $ClO_2^-$, the equilibrium concentration of $ClO_2\cdot$ is $6[I_2]$, whereas $ClO_2\cdot$ is $0.2[ClO_2^-]$ with excess amount of $I_2$[50]. Assuming that there is enough amount of the $I_2$, the concentration of $ClO_2\cdot$ can be estimated as $0.2[ClO_2^-] = 2\times10^{-3}M$ in our experiments. Similar to $ClO_2\cdot$, the solutions of the reagents do not contain $I_2$, but it forms in the following chemical reaction:

$$IO_3^- + 5I^- + 6H^+ \rightarrow 3I_2 + 3H_2O \tag{20}$$

In our experiments, the solution contains an excess of $IO_3^-$. Therefore, the concentration of $I_2$ is proportional to the concentration of $I^-$ (which is $2\times10^{-3}$ M), and it is $1.2\times10^{-3}$ M based on the stoichiometry of the reaction (Eq. (20)). A part of $I_2$ is consumed by the chlorite–iodine reaction. The consumption of $I_2$ can be calculated as $\frac{1}{6}[ClO_2\cdot]$ based on the measurements by Rábai and Beck[50]. Thus, the concentration of the $I_2$ can be estimated as $1.2\times10^{-3} - 3.3\times10^{-4} = 8.7\times10^{-4}$ M. From these estimated concentrations, the obtained values are the following: $a = \frac{k_1[MA]_0}{k_2[ClO_2\cdot]_0\sqrt{\alpha}} = 5.9\times10^3[MA]_0$ and $b = \frac{k_3[I_2]_0}{k_2[ClO_2\cdot]_0\sqrt{\alpha}} = 1.9$.

**Numerical model**. The RD equations (Eqs. (1) and (2)) were solved numerically using a method of lines technique with the finite difference method on a uniform 2D square grid (with various grid spacings $1.25\times10^{-1}$ and $2.5\times10^{-1}$). In the spatial discretization, a second-order five-point stencil was used with the alternating direction implicit method (ADI). ADI is an operator splitting method, namely the explicit and the implicit methods were applied alternately in the space directions. The implicit space derivative was calculated with the lower–upper (LU) decomposition. The reaction kinetic term was calculated with the second-order Heun's method. In the numerical integration, we used various time steps of $3.124\times10^{-4}$, $6.25\times10^{-4}$, and $1.3\times10^{-3}$. The application of the grid spacing and time step (indicated in the figure captions) depended on the problem, but it did not quantitatively affect the generated patterns. We applied the following initial conditions $u(t = 0, x, y) = 0$ and $v(t = 0, x, y) = 0$, to which noise was added, having a normal distribution with a mean and a standard deviation of 0 and 0.01, respectively. In the simulations, only the parameter a was varied (constant or sinusoidally modulated), and the values of all other parameters were fixed: $\sigma = 8$, $b = 1.9$ (estimated from the experimental data), and $c = 1.5$. We applied periodic boundary conditions for u and v. All parameters and variables in the model are dimensionless.

**Linear stability analysis**. First, we address the linear stability analysis of uniform periodic solutions to the model (Eqs. (1) and (2)). Let $u_p(t)$ and $v_p(t)$ designate uniform periodic solutions to the model equation without the diffusion terms

$$\frac{du}{dt} = a + A\sin\frac{2\pi}{T}t - u - \frac{4uv}{1 + u^2} \tag{21}$$

$$\frac{dv}{dt} = \sigma\left(b\left(u - \frac{uv}{1 + u^2}\right)\right) \tag{22}$$

We note that the convergence to periodic solutions depends on the values of the parameters and that such functions satisfy $u_p(t) = u_p(t + T)$ and $v_p(t) = v_p(t + T)$. Considering the linear stability analysis of the periodic solutions with respect to the partial differential equations (Eqs. (1) and (2)), we perturb as follows:

$$u(t, x) = u_p(t) + \delta u^{(k)}(t)e^{-ikx} \tag{23}$$

$$v(t, x) = v_p(t) + \delta v^{(k)}(t)e^{-ikx} \tag{24}$$

where k is a real wavenumber. Substituting these expressions into Eq. (1), we obtain

$$\frac{\partial}{\partial t}\left(u_p(t) + \delta u^{(k)}(t)e^{-ikx}\right) = a + A\sin\frac{2\pi}{T}t - u_p(t) + \delta u^{(k)}(t)e^{-ikx}$$
$$- \frac{(4u_p(t) + \delta u^{(k)}(t)e^{-ikx})(v_p(t) + \delta v^{(k)}(t)e^{-ikx})}{1 + \left(u_p(t) + \delta u^{(k)}(t)e^{-ikx}\right)^2}$$
$$+ \nabla^2\left(\delta u^{(k)}(t)e^{-ikx}\right) \tag{25}$$

Noting that the periodic solution satisfies

$$\frac{\partial u_p(t)}{\partial t} = a + A\sin\frac{2\pi}{T}t - u_p(t) - \frac{4u_p(t)v_p(t)}{1 + u_p(t)^2} \tag{26}$$

and neglecting terms with order higher than $\delta u^{(k)}$ and $\delta v^{(k)}$, we obtain

$$\frac{d\delta u^{(k)}(t)}{dt} = \left[-1 + \frac{4v_p(t)\left(u_p(t)^2 - 1\right)}{\left(u_p(t)^2 + 1\right)^2} - k^2\right]\delta u^{(k)}(t) - \frac{4u_p(t)}{u_p(t)^2 + 1}\delta v^{(k)}(t) \tag{27}$$

In the same manner, the equation satisfied by $\delta v^{(k)}(t)$ becomes

$$\frac{d\delta v^{(k)}(t)}{dt} = \sigma b\left[1 + \frac{v_p(t)\left(u_p(t)^2 - 1\right)}{\left(u_p(t)^2 + 1\right)^2}\right]\delta u^{(k)}(t) - \sigma\left[b\frac{u_p(t)}{u_p(t)^2 + 1} + ck^2\right]\delta v^{(k)}(t) \tag{28}$$

By defining the time-dependent matrix $L(t;k)$ as

$$\mathbf{L}(t;k) = \begin{pmatrix} -1 + \frac{4v_p(t)(u_p(t)^2-1)}{(u_p(t)^2+1)^2} - k^2 & -\frac{4u_p(t)}{u_p(t)^2+1} \\ \sigma b\left[1 + \frac{v_p(t)(u_p(t)^2-1)}{(u_p(t)^2+1)^2}\right] & -\sigma\left[b\frac{u_p(t)}{u_p(t)^2+1} + ck^2\right] \end{pmatrix} \tag{29}$$

we are able to express Eqs. (27) and (28) as the following differential equation:

$$\frac{d}{dt}{}^t\left(\delta u^{(k)}(t), \delta v^{(k)}(t)\right) = \mathbf{L}(t;k){}^t\left(\delta u^{(k)}(t), \delta v^{(k)}(t)\right) \tag{30}$$

The analysis of Eq. (30) enables one to determine the stability of the periodic solutions. To this end, we have employed a semi-analytical approach. In particular, for a large natural number N, approximate periodic solutions are constructed using Euler's method with a time step $\triangle t = T/N$. In order to analyze the stability, the time evolution of $\delta u^{(k)}$ and $\delta v^{(k)}$ are obtained during one period of the external

forcing. For given initial conditions $\delta u^{(k)}(0)$ and $\delta v^{(k)}(0)$, we obtain an approximation of the Poincaré map as follows:

$$^{t}(\delta u^{(k)}(T), \delta v^{(k)}(T))$$

$$= \hat{\mathbf{L}}\left(\frac{N-1}{N}T; k\right)\hat{\mathbf{L}}\left(\frac{N-2}{N}T; k\right)\cdots\hat{\mathbf{L}}\left(\frac{2}{N}T; k\right)\hat{\mathbf{L}}\left(\frac{1}{N}T; k\right)\hat{\mathbf{L}}(0; k)^{t}(\delta u^{(k)}(0), \delta v^{(k)}(0))$$

$$\equiv \mathbf{P}(k)^{t}(\delta u^{(k)}(0), \delta v^{(k)}(0)) \tag{31}$$

where $\hat{\mathbf{L}}(s; k) = (\mathbf{I} + \triangle t \mathbf{L}(s; k))$, and $\mathbf{I}$ is the unit matrix. Equation (31) is considered as an autonomous time-discrete dynamical system for the perturbation added to the uniform oscillatory state with the spatial wavenumber $k$. In other words, $\mathbf{P}(k)$ is a linearized Poincaré map for the perturbation with the wavenumber $k$. Denoting the eigenvalues of $\mathbf{P}(k)$ as $\lambda_1(k)$ and $\lambda_2(k)$, it follows that if $|\lambda_1(k)|$ and $|\lambda_2(k)|$ both less than one for all $k$, then the uniform periodic solution without the spatial gradient is stable. If, on the other hand, the values of either of $|\lambda_1(k)|$ and $|\lambda_2(k)|$ exceed one for a range of $k$, then the uniform periodic solution is unstable.

## Data availability
The data that support the findings of this study are available from the corresponding authors upon reasonable request. Supplementary figures included in the Supplementary information document, and Supplementary Movies 1, 2, 3, 4, 5, 6, and 7 are available.

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

## Acknowledgements
This work was supported by the National Research, Development and Innovation Office of Hungary (K131425 and K134687), the National Research, Development, and Innovation Fund of Hungary under Grant TKP2021-EGA-02, and JSPS Japan–Hungary Bilateral Joint Research Project (JPJSBP 120213801).

## Author contributions

I.L. and I.S. conceived the study. B.D. and I.S. designed the gel reactor. B.D. carried out the experiments. G.H. performed the numerical simulations of the RD equations. H.K., E.G., and N.J.S. performed the stability analysis and estimation of the parameters of the numerical model from the experiential data. All authors interpreted the data. The manuscript was written through the contributions of all authors.

## Competing interests

The authors declare no competing interests.

## Additional information

 

