## [Peer Review File · Communications Chemistry]

Reviewers' comments:

Reviewer #1 (Remarks to the Author):

The authors propose a mechanism for controlling Turing patterns via a time-varying inflow of one of the two chemical species. They give both experimental and theoretical results (the latter from numerical simulations of a reaction-diffusion system). I find the ideas interesting and worth publishing, as they provide a new method for real-time control of Turing patterns in chemical systems.

I have relatively minor comments which should be considered by the authors. Specific comments include:

1. In the theoretical method (1)-(2), the time-varying parameter enters into the first PDE (1). Yet, from the experimental discussion, it seems that the forcing (e.g., the time-varying inflow) occurs at the boundary. Why not have the forcing in a boundary condition, instead of in the PDE? This modelling choice should be better discussed.

2. What is the space domain of the PDE? Is it a 2D square? More specifics on this are needed. What are the boundary conditions? What is the level of error in the numerical simulations?

3. For both experiments and theory: The figures mostly look blurry/low resolution, and this needs to be improved. I found some of the images hard to read because of it. I suggest using high resolution vector graphics or png, but never use jpg and the quality is too low.

4. There are separate figures for experiments and theory (simulations). Is there any common data that can be plotted and compared so that we can see how well the theory can explain the experiments? For instance, a more specific comparison of plots like Fig 2.(bi,ci,di) with Fig. 3(ai,bi)? Some comparison to connect theory to experiment would benefit the paper.

5. Regarding the instability analysis starting at line 370, it may be possible to better understand the nature of instabilities analytically without directly resorting to a simulation. The authors earlier mention the paper [35] as one manner of time-varying control at the boundary (in that case, temperature). In that paper, a completely analytical method for the instability analysis was used, which was earlier proposed another paper by the same author:

<https://doi.org/10.1098/rspa.2020.0003>

It seems like the stability problem (29) in the present paper can be solved by the method in that paper. In case this is too much work, I will just mention it as a possibility that the authors should consider going forward in their future work.

Reviewer #2 (Remarks to the Author):

The paper by Duzs et al. presents the influence of periodic changes in feed on Turing pattern formation. They experimentally found interesting phenomena that Turing patterns can be formed by oscillatory feeding and patterns can be switched after the oscillation. They reproduced them by simulations and explained by linear stability analysis. The experiments and analyses were carefully performed in the presented conditions. However, the reason for choosing this oscillation frequency is not explained at all. The frequency dependence needs to be addressed to decide to recommend this paper for publication.

In the extremely fast input oscillation, the reaction-diffusion system likely only responds to the

average input. In the opposite limit, the system completely follows the pattern as in the constant input condition. Thus, the important condition is the ratio of the oscillation period and the system relaxation time. Here, the movies show the input oscillation is faster than the relaxation of Turing patterns. However, no explanation is given in the text. The dependence on the oscillation frequency should be investigated in detail. Particularly, the frequency can be easily varied in the simulation.

The bistability of patterns should be quantitatively discussed. For pattern switching, it is a fundamental property. In the text, It is only mentioned that it is difficult to observe in experiments. In the present experimental setup, the authors can gradually increase or decrease the feed concentration. I think it is a suitable setup to investigate the bistability and hysteresis. Also, the bistability should be addressed in simulation.

Reviewer #3 (Remarks to the Author):

The authors present results of the effects of periodic forcing on Turing patterns. There is great interest in mechanisms of control of the Turing pattern forming mechanism and to my knowledge, this has not been achieved before by periodic variations in feed concentration; the majority of work in the area is performed using light. The authors find that patterns can be suppressed or initiated by periodic changes in concentrations, which does suggest an alternative method for controlling spatiotemporal pattern selection. However, the discussion and conclusions could more clearly articulated.

There is discussion of robustness "patterns must survive some variation in the environmental conditions". This would generally be understood as the patterns survive when the environmental conditions are changed – as investigated in a number of studies eg the paper cited by Vittadello et al "a model's predictions remain largely unchanged when aspects of the model—parameters, details of the model structure, external influences—are changed". The fact that patterns disappear during the forcing could be argued that they are not robust to the environmental changes. It is generally expected that the same patterns return after all environmental changes are removed and the system is set at exactly the same initial parameters - either there is a single state that the system must return to i.e. spots, or the system is bistable in which case there is hysteresis, so the system may return to the original state, depending on the magnitude of the perturbation.

However, the concentrations for Figure 1b are $MA = 7 \text{ mM}$ and looking at Figure S1, it appears that there is bistability between a mixed spot/stripe state and waves at $MA = 7 \text{ mM}$, rather than hexagonal spots? Probably there is slight variation in the position of the boundaries between states from experiment to experiment but there is no discussion of this on Pg 4 or alongside S1 - this is a very important point when discussing robustness. Such bistability is not shown in the simulations in Figure S5.

Looking at the movie for Figure 1b, there are large amplitude but uniform changes in the colour of the domain during the forcing that are not mentioned in the main text. It is described in Figure S12 as a "slight colour change in tank B due to the periodic forcing". In the simulations it appears also that there are oscillations in u along with the forcing. I am not sure this well described as the "time-independent pattern changes" as depicted in figure 1, since there is a transition from a spatial pattern with no time-dependency to no spatial pattern with a time-dependent state. Similar for 1c.

Figure 1d looks odd – there is a dark band in the image 1 and the patterns only appear in the bottom half of the image. Looking at the movie, there is a lot of shadow. Perhaps an experimental run where there was an issue with the stirrer bar and hence there is a gradient? Are there any better examples of this?

If the aim of the paper is really centred around demonstrating robustness, I would be expecting to see many experimental runs and statistics on the number of times a particular state is observed for both the stationary inflow and periodic inflow. It would be important to determine the effect of different

amplitudes and period of forcing with return to the same initial MA. But to my mind, the discussion should be more focused on spatiotemporal pattern selection/ control than robustness – in which case the results demonstrate how periodic changes in feed can be used.

Minor points

In the section beginning numerical simulations in the main text, it would be useful to say what u and v correspond to chemically.

There could be more discussion on the insights gained from the linear stability analysis in comparison to the numerical simulations.

Some typical volumetric flow rates for q_1 , q_2 etc would be useful in the experimental methods section.

We thank the Reviewers for the positive assessment of our work and for the valuable comments. We hope that the replies we provide below address adequately the criticized points.

Reviewer #1:

“The authors propose a mechanism for controlling Turing patterns via a time-varying inflow of one of the two chemical species. They give both experimental and theoretical results (the latter from numerical simulations of a reaction-diffusion system). I find the ideas interesting and worth publishing, as they provide a new method for real-time control of Turing patterns in chemical systems.”

Author reply: We thank the Reviewer for his/her comments and the positive assessment of our work.

“I have relatively minor comments which should be considered by the authors. Specific comments include:

1. In the theoretical method (1)-(2), the time-varying parameter enters into the first PDE (1). Yet, from the experimental discussion, it seems that the forcing (e.g., the time-varying inflow) occurs at at the boundary. Why not have the forcing in a boundary condition, instead of in the PDE? This modelling choice should be better discussed.”

Author reply: In a TSFR, Turing patterns form in a layer perpendicular to gradients imposed by the boundary conditions. The size of our reactor in the perpendicular dimensions is significantly larger ($d_p = 20$ mm) than in the direction of the gradient ($d_g = 2.5$ mm). The effect of the gradient in forming Turing patterns in the CIMA reaction has been studied experimentally [Dulos 1996, Rudovics 1996] and theoretically.[Boissonade 1988, Lengyel 1995] These studies pointed out that in the inherently three-dimensional system, the strong anisotropy forms quasi-two-dimensional patterns. Therefore, reduced two-dimensional simulations show reasonable qualitative agreement with the experimental observations.[Lengyel 1991] This does not mean that a three-dimensional effect cannot occur under some conditions, as numerical simulations made in parameter gradients show the possibility of forming different patterns at different locations along the gradients.[Borckmans 1992, Setayeshgar 1998] We have not observed clear signs of three-dimensional effects in our experiments. Therefore, we used reduced two-dimensional modeling, where the feeding from the boundary is described by parameters that appear in the PDE.

The other issue that must be considered is the level of description of the actual chemistry. Quantitative modeling of the CIMA reaction could be made by the nine-variable model of Lengyel and coworkers [Lengyel 1996] or at least by a five-variable version.[Lengyel 1991] A chemically

reasonable simplification of these detailed models to a two-variable one was proposed by Lengyel and Epstein.[Lengyel 1991] This model shows the structure of a general activator-inhibitor system and allows the numerical and analytical study of pattern formation. We aimed to apply this simple model to understand the general aspects of the experimental observations, which are not determined by the particular chemistry of the reaction.

Two new paragraphs are added to discuss the modeling choice on page 7.

“2. What is the space domain of the PDE? Is it a 2D square? More specifics on this are needed. What are the boundary conditions? What is the level of error in the numerical simulations?”

Author reply: The PDE was solved on a 2D square domain with 401×401 grid points (and in the case of the phase diagrams with 201×201 grid points) with the finite difference method. In the spatial discretization, a second-order five-point stencil was used with the alternating direction implicit method (ADI). ADI is an operator splitting method: the explicit and the implicit methods were applied alternately in the space directions. The implicit space derivative was calculated with the LU decomposition so the number of floating point operations was similar in the case of the implicit and in the case of the explicit direction as well. The ADI method is stable for any time step (in parabolic differential equations) like the Crank-Nicholson method, but computationally it is more effective. On the boundary periodic boundary conditions were applied, however, the no flux boundary condition was tried out as well, and it gave (qualitatively) the same result. The reaction kinetic term was treated with the second order Heun's method. We rewrote the corresponding part in the text (Numerical model).

“3. For both experiments and theory: The figures mostly look blurry/low resolution, and this needs to be improved. I found some of the images hard to read because of it. I suggest using high resolution vector graphics or png, but never use jpg and the quality is too low.”

Author reply: We appreciate this comment, and we tried to improve the quality of the figures. The original figures already have better quality, but possibly we lost it by the pdf conversion.

“4. There are separate figures for experiments and theory (simulations). Is there any common data that can be plotted and compared so that we can see how well the theory can explain the experiments? For instance, a more specific comparison of plots like Fig 2.(bi,ci,di) with Fig. 3(ai,bi)? Some comparison to connect theory to experiment would benefit the paper.”

Author reply: In the experiments, we observe only the color change of the indicator, which corresponds to the triiodide concentration, but from the recorded pictures, we cannot calculate this concentration. The model variables are the dimensionless concentration of iodide and chlorite ions. We can do only the qualitative comparison of the grayscale images and calculated values of the variables. The aim of modeling is not to explain the experiments quantitatively but to clarify the general aspects of the experimental observations. A quantitative comparison would require using a more specific kinetic model, but this was not our aim in this paper.

“5. Regarding the instability analysis starting at line 370, it may be possible to better understand the nature of instabilities analytically without directly resorting to a simulation. The authors earlier mention the paper [35] as one manner of time-varying control at the boundary (in that case, temperature). In that paper, a completely analytical method for the instability analysis was used, which was earlier proposed another paper by the same author: <https://doi.org/10.1098/rspa.2020.0003> It seems like the stability problem (29) in the present paper can be solved by the method in that paper. In case this is too much work, I will just mention it as a possibility that the authors should consider going forward in their future work.”

Author reply: We have evaluated the instability based on the discussion in the paper by van Gorder (Proc. R. Soc. A.47620200003, 2020). The condition shown in Eq. (12) in the paper by van Gorder is a sufficient condition for the instability for the perturbation with wavenumber k . We have numerically evaluated the inequality in Eq. (12) and found that it does not hold for a certain time within a period at the wavenumber $k \approx 1.0$ for both $A = 1$ and 2 . It should be noted that the uniform state is stable and unstable for $A = 1$ and 2 , respectively. Considering that, the theorem claims that if the inequality holds for all time then the system becomes unstable for the perturbation with a wavenumber k . Therefore, the theorem in the paper by van Gorder cannot determine the boundary of the instability in our system.

Figure R1: Plot of the left-hand side subtracted from right-hand side in the inequality (5.12) in the reference against time for each wavenumber k . The parameters are set as $a = 22$, $\sigma = 8$, $b = 1.9$, $c = 1.5$, $T = 1$, and $N = 10000$. (a) $A = 1$. (b) $A = 2$. In both the cases inequality (5.12) does not hold for certain ranges in t , which means that inequality (5.12) cannot conclude whether the system is stable or not.

Reviewer #2

“The paper by Duzs et al. presents the influence of periodic changes in feed on Turing pattern formation. They experimentally found interesting phenomena that Turing patterns can be formed by oscillatory feeding and patterns can be switched after the oscillation. They reproduced them by simulations and explained by linear stability analysis. The experiments and analyses were carefully performed in the presented conditions. However, the reason for choosing this oscillation frequency is not explained at all. The frequency dependence needs to be addressed to decide to recommend this paper for publication. In the extremely fast input oscillation, the reaction-diffusion system likely only responds to the average input. In the opposite limit, the system completely follows the pattern as in the constant input condition. Thus, the important condition is the ratio of the oscillation period and the system relaxation time. Here, the movies show the input oscillation is faster than the relaxation of Turing patterns. However, no explanation is given in the text. The dependence on the oscillation frequency should be investigated in detail. Particularly, the frequency can be easily varied in the simulation.”

Author reply: We thank the Reviewer for his/her comments. This comment points out a significant issue. We have previously addressed the effect of changing the amplitude and the frequency of forcing numerically in the case of spatiotemporal oscillations in TSFR.[Dúzs 2021] We found that low-frequency forcing results in a complete periodic excursion through the different states of the system. The increase in frequency leads to forced bursting phenomena. While at high-frequency forcing, the frequency of the spatiotemporal oscillations adjusts to that of the forcing. We have used these results to guide our experiments on Turing patterns and checked different experimental conditions (amplitude and frequency). To find conditions where the forcing is adequate, we have considered the following points:

- (1) The forcing frequency must fit the inherent frequency of the unperturbed spatiotemporal oscillations. Swinney and coworkers found that the typical period of oscillations near a Hopf bifurcation is between 9 and 60 s in the CIMA reaction.[Ouyang 1995]
- (2) The forcing frequency must fit the response time of the system to variations in the boundary conditions. According to our and others' experiences [Lengyel 1995], the patterns in the CIMA reactions respond relatively rapidly, within a few minutes, to variations in the boundary conditions. However, the appearance and disappearance of the Turing patterns (i.e., the transition between pattern and no pattern states) take longer (30-60 min).

- (3) The tank damps the forcing, which is significant when the forcing period is shorter than the residence of the stirred tank reactor. [Dúzs 2021] In the experiments, the forcing is applied to the tank feeding, and the composition of the mixture in the tank sets the boundary condition for the gel part. As the residence time of the tank is 2-3 min, the period of the forcing must exceed it.

Considering all this information, we choose a forcing period of 5 and 10 minutes. This is discussed on page 5 in the revised manuscript.

We performed numerical and semi-analytical studies to clarify the effect of the forcing frequency and amplitude. The latter shows that the stability of the uniform oscillatory state behaves similarly to what is often observed in resonance-induced instabilities. In particular, for specific amplitudes A , the length of the period T can be varied to traverse from regions of stability to those of instability. A new figure (Figure 5) is introduced in the manuscript. The numerical results are in agreement with these results (Figure S7).

“The bistability of patterns should be quantitatively discussed. For pattern switching, it is a fundamental property. In the text, It is only mentioned that it is difficult to observe in experiments. In the present experimental setup, the authors can gradually increase or decrease the feed concentration. I think it is a suitable setup to investigate the bistability and hysteresis. Also, the bistability should be addressed in simulation.”

Author reply: We have observed bistability in the experiments between the homogeneous no pattern state and Turing patterns, and between the Turing patterns and spatiotemporal waves. The corresponding figure in the supporting file has been modified to present it more clearly (Figure S1.

The original sentence (“As the transition from the homogeneous state to the spot patterned state is generically subcritical, bistability between the homogeneous and the patterned state is expected, but it is difficult to observe it experimentally.”) could be misunderstood. Checking the stability limits of the coexisting states requires long-lasting experiments where the conditions must be kept rigorously. That can be difficult, e.g., to avoid any changes in the composition of the feeding solutions, temperature, etc. Also, if the regime of bistability is narrow, the determination of its boundaries is also challenging. We made additional simulations to demonstrate this bistability between the homogeneous and patterned states clearly (Figure S5 and S6). The appearance of bistability is discussed on page 4 in the revised manuscript.

Reviewer #3

“The authors present results of the effects of periodic forcing on Turing patterns. There is great interest in mechanisms of control of the Turing pattern forming mechanism and to my knowledge, this has not been achieved before by periodic variations in feed concentration; the majority of work in the area is performed using light. The authors find that patterns can be suppressed or initiated by periodic changes in concentrations, which does suggest an alternative method for controlling spatiotemporal pattern selection. However, the discussion and conclusions could more clearly articulated. “

Author reply: We thank the Reviewer for his/her comments.

“There is discussion of robustness “patterns must survive some variation in the environmental conditions”. This would generally be understood as the patterns survive when the environmental conditions are changed – as investigated in a number of studies eg the paper cited by Vittadello et al “a model’s predictions remain largely unchanged when aspects of the model—parameters, details of the model structure, external influences—are changed”. The fact that patterns disappear during the forcing could be argued that they are not robust to the environmental changes. It is generally expected that the same patterns return after all environmental changes are removed and the system is set at exactly the same initial parameters - either there is a single state that the system must return to i.e. spots, or the system is bistable in which case there is hysteresis, so the system may return to the original state, depending on the magnitude of the perturbation. However, the concentrations for Figure 1b are $MA = 7$ mM and looking at Figure S1, it appears that there is bistability between a mixed spot/stripe state and waves at $MA = 7$ mM, rather than hexagonal spots? Probably there is slight variation in the position of the boundaries between states from experiment to experiment but there is no discussion of this on Pg 4 or alongside S1 - this is a very important point when discussing robustness. Such bistability is not shown in the simulations in Figure S5.”

Author reply: In the experiments, we observed bistability between the homogeneous and patterned states and between waves and patterned states, but we could not see that between the hexagonal spots and stripes. In the domain of the theoretically supported hysteresis between hexagons and stripes, metastability determines which structure is favored. Minor disturbances, inhomogeneities in the gel, and especially slight variation of the gel thickness (the size of the gel in the direction of the gradients in a TSFR) make the experimental observation of this sort of bistability unlikely,[Borckmans 2022] and leads to the formation of a mixed pattern (hexagon and stipe). Therefore, our experiments could not answer the problem of pattern selection. We can observe the switch between waves, homogeneous states, and pattern (spots and/or stripes) states. We have performed numerical simulations to support the experimentally observed bistability phenomenon.

“Looking at the movie for Figure 1b, there are large amplitude but uniform changes in the colour of the domain during the forcing that are not mentioned in the main text. It is described in Figure S12 as a “slight colour change in tank B due to the periodic forcing”.

Author reply: The camera looked through the two tanks and the gel part in our experimental setup. The solution in Tank A is colorless, but tank B is reddish due to the small amount of PVA-I₃ complex. The forcing generally results in a periodic slight color change in tank B that can be recognized in the movies. When the contrast of the pictures was enhanced (Fig. 2d), we also see the additional color changes in tank B (not only during the forcing, but during the whole experiment) due to the not completely perfect mixing around the inlet of solutions in Tank B. This effect appeared in the corresponding pictures as a superposition but did not result in any distortions or unwanted gradients in the gel where patterns formed.

“In the simulations it appears also that there are oscillations in u along with the forcing. I am not sure this well described as the “time-independent pattern changes” as depicted in figure 1, since there is a transition from a spatial pattern with no time-dependency to no spatial pattern with a time-dependent state. Similar for 1c.

Figure 1d looks odd – there is a dark band in the image 1 and the patterns only appear in the bottom half of the image. Looking at the movie, there is a lot of shadow. Perhaps an experimental run where there was an issue with the stirrer bar and hence there is a gradient? Are there any better examples of this?”

Author reply: We agree that the “time-independent pattern changes” can be misunderstood. We rewrite the caption of Figure 1 as: “Due to the continuous periodic forcing of the input feed concentration of one reactant, the sustained pattern changes drastically (right column).” Unfortunately, we have no better experimental recordings to replace Figure 1d.

“If the aim of the paper is really centred around demonstrating robustness, I would be expecting to see many experimental runs and statistics on the number of times a particular state is observed for both the stationary inflow and periodic inflow. It would be important to determine the effect of different amplitudes and period of forcing with return to the same initial MA. But to my mind, the discussion should be more focused on spatiotemporal pattern selection/ control than robustness – in which case the results demonstrate how periodic changes in feed can be used.”

Author reply: The experimental observations are well-reproducible within the accuracy of this type of experiment. Our findings in the absence of forcing agree with the experiments performed by the Bordeaux group more than 25 years ago.[Rudovics 1996] We repeated our experiments several times and observed the same results within a few tens of mM variations in the input feed concentrations. We did not intend to use robustness here as a statistical characteristic. Following this comment, we replace “robustness” with “resilience,” as we want to show that patterns arising from Turing instability can survive significant changes in environmental parameters. The actual experiments cannot provide enough information to discuss the problem of pattern selection.

“Minor points

In the section beginning numerical simulations in the main text, it would be useful to say what u and v correspond to chemically. “

Author reply: It was a mistake to skip the definition of u (activator, iodide) and v (inhibitor, chlorite). We corrected it.

“There could be more discussion on the insights gained from the linear stability analysis in comparison to the numerical simulations. “

Author reply: In a standard stability analysis, we consider the stationary uniform state and growth rates for the perturbation with a certain wave number are then calculated. In contrast, here the growth rates for the perturbation of certain wave numbers are calculated with respect to the uniform oscillatory state (i.e., it is no longer stationary). Therefore, to evaluate growth rates, we must define a map for the time evolution over its period, and then determine the time evolution of the perturbation as a map.

The advantage of the present method is that we can discuss the stability of the system without directly calculating the PDE. This allows us to vary the parameters and investigate the mathematical structure of the instability.

By varying the amplitude and frequency, we see that the instability of the oscillatory uniform states under periodic forcing can easily occur within certain frequency ranges. The dependence is nonlinear and this may reflect the intrinsic mathematical structure of the system.

“Some typical volumetric flow rates for q_1 , q_2 etc would be useful in the experimental methods section.”

Author reply: We added the following sentences to the experimental section:

In our experiments the volume of each tank was 26 mL, $q_2 = 499$ mL/h, and the sinusoidal forcing was made with q_0 varied in the range of 90-170 mL/h. In case of lower or higher desired [MA] concentrations, we used a 20 mM or 45 mM MA stock solution, respectively. With this, we could keep the q_0 in the abovementioned optimal range. The A of the modulation was varied between 50-170 mL/h.

References

Dulos, E., Davies, P., Rudovics, B., & De Kepper, P. (1996). From quasi-2D to 3D Turing patterns in ramped systems. *Physica D: Nonlinear Phenomena*, 98(1), 53-66.

Rudovics, B., Dulos, E., & De Kepper, P. (1996). Standard and nonstandard Turing patterns and waves in the CIMA reaction. *Physica Scripta*, 1996(T67), 43.

J. Boissonade. Stationary structure induced along a reaction-diffusion front by a Turing symmetry breaking instability. *Journal de Physique*, 1988, 49 (3), pp.541-546.

Lengyel, I., & Epstein, I. R. (1995). The chemistry behind the first experimental chemical examples of Turing patterns. In *Chemical Waves and Patterns* (pp. 297-322). Springer, Dordrecht.

Lengyel, I., & Epstein, I. R. (1991). Modeling of turing structures in the chlorite—iodide—malonic acid—starch reaction system. *Science*, 251(4994), 650-652.

Borckmans, P., De Wit, A., & Dewel, G. (1992). Competition in ramped Turing structures. *Physica A: Statistical Mechanics and Its Applications*, 188(1-3), 137-157.

Setayeshgar, S., & Cross, M. C. (1998). Turing instability in a boundary-fed system. *Physical Review E*, 58(4), 4485.

Lengyel, I., Li, J., Kustin, K., & Epstein, I. R. (1996). Rate constants for reactions between iodine- and chlorine-containing species: a detailed mechanism of the chlorine dioxide/chlorite-iodide reaction. *Journal of the American Chemical Society*, 118(15), 3708-3719.

Dúzs, B., Molnár, I., Lagzi, I., & Szalai, I. (2021). Reaction–Diffusion Dynamics of pH Oscillators in Oscillatory Forced Open Spatial Reactors. *ACS omega*, 6(50), 34367-34374.

Ouyang, Q., Li, R., Li, G., & Swinney, H. L. (1995). Dependence of Turing pattern wavelength on diffusion rate. *The Journal of chemical physics*, 102(6), 2551-2555.

Borckmans, P., Dewel, G., De Wit, A., Dulos, E., Boissonade, J., Gauffre, F., & De Kepper, P. (2002). Diffusive instabilities and chemical reactions. *International journal of bifurcation and chaos*, 12(11), 2307-2332.

Reviewers' comments:

Reviewer #1 (Remarks to the Author):

I find most of my comments were adequately addressed, except for one.

I did not find the comparison with the instability theorem of [50] was quite correct. The authors state that the inequality of [50] does not hold for all time, so they should not use it. However, that was the point of my suggestion - that the stability/instability should hold for some times, not all times (e.g., over some time interval I_k , where each I_k will be different for the k -th Turing mode). This tells one when a Turing instability will be present, leading to patterns. When the instability is suppressed, the inequality changes sign, and then the solutions instead tend to a spatially homogeneous state.

In the present paper, the authors find that their system evolves between a patterned state and a spatially homogeneous state (as shown in some figures and movies). This means that each state must gain and lose stability over different time intervals. So, no Turing mode should be unstable for all time under such conditions. In particular, perturbations of the $k=0$ state against Turing modes for $k \geq 1$ should give instabilities when the dynamics tend to the patterned state, while when the dynamics are spatially uniform all these perturbations should be stable. This means that the inequality in [50] corresponding to such a system will change sign.

Now, if we compare this to the semi-analytic approach of the authors, we see that the authors always find an instability result which is independent of time. This would suggest a pattern which emerges and then persists for all time, yet this is not what the numerical simulations show. Let us look, for example, at Figure 3a(ii-iv). In this figure, we see that the oscillatory motion suppresses the Turing instability (panel iii), and after it is switched off the Turing instability appears again (panel iv). The instability analysis of the authors would not predict this absence of patterns for the region where oscillation are switched on, since it neglects time.

In their response, the authors state:

"The condition shown in Eq. (12) in the paper by van Gorder is a sufficient condition for the instability for the perturbation with wavenumber k . We have numerically evaluated the inequality in Eq. (12) and found that it does not hold for a certain time within a period at the wavenumber $k \approx 1.0$ for both $A = 1$ and 2 . It should be noted that the uniform state is stable and unstable for $A = 1$ and 2 , respectively. Considering that, the theorem claims that if the inequality holds for all time then the system becomes unstable for the perturbation with a wavenumber k . Therefore, the theorem in the paper by van Gorder cannot determine the boundary of the instability in our system."

They also state:

"In both the cases inequality (5.12) does not hold for certain ranges in t , which means that inequality (5.12) cannot conclude whether the system is stable or not."

The mistake of the authors is they assume that the boundary of the stability region is fixed over time. In contrast, the stability region itself must vary in time, in order to generate the alternation between the presence and absence of patterns as seen in fig. 3(a) and some of the movies.

Indeed, the results discussed in [50] only hold for certain ranges of time, rather than all time, by design. This allows one to consider the subset of times for which a given Turing mode is unstable, and hence to understand patterns which are only present some and not all of the time, just like the patterns in the paper under review. Looking at the examples in [50], we see that a change in the

instability region tends to correspond to a change in the pattern present over time. This was also true in the paper [35], where a change in boundary temperature over time resulted in a change in the instability region and hence of the Turing patterns over time. See Figures 11, 12, and 13 in that paper - all of which feature time-varying instability regions corresponding to Turing patterns which change in time.

Part of the reason the authors may be confused is that they are seeking temporal eigenvalues in order to determine stability/instability of the Turing modes. However, it is well known that eigenvalues are no longer useful indicators of stability for systems with time-varying parameters; see, for instance, <http://www.chebfun.org/examples/ode-linear/FrozenCoeffs.html>

O. Perron, Die Stabilitätsfrage bei Differentialgleichungen, Math. Zeit. 32 (1930), 703-728.

M. Wu, A note on stability of linear time-varying systems, IEEE Trans. Autom. Control 19 (1974), 162-162.

Josić K, Rosenbaum R. 2008 Unstable solutions of nonautonomous linear differential equations. SIAM Rev. 50, 570-584.

I would suggest the authors think a bit more about these points regarding the stability analysis. As it stands, the results of their stability analysis stand in contrast to the results of their numerical simulations.

Reviewer #2 (Remarks to the Author):

The authors have responded to my comments well. They have conducted additional simulations and stability analysis to clarify the bistability and oscillation frequency dependence. I recommend this manuscript for publication if the following minor points are addressed.

The coexistence of two types of Turing patterns is shown in Fig. S6, whereas a homogeneous oscillation phase is shown above $a = 27$ in Fig. S5. Is this difference due to hysteresis? Do the Turing and oscillation phases coexist at $27 < a < 40$? It should be specified. The explanation in the text is not sufficient. In Page 7, the sixth line from Eq.(2), "We found that the Turing pattern formation occurs between $a=23$ and $a=27$ from spots to stripes. Below $a=23$ and above $a=27$, stationary homogeneous patterns and homogeneous oscillations exist, respectively". This is only for a specific initial condition.

A larger spacing grid and longer Delta t are used for the simulations for new figures. If they do not change the results, it is OK but should be mentioned somewhere.

It is written that the initial state of Fig. S7(b) is a Turing state. It is probably a spot pattern. The authors should specify it more exactly.

The section of the Numerical model in Page 15 should be updated. Different spacing, time steps, and initial conditions are also used in the additional simulations.

Reviewer #3 (Remarks to the Author):

I am happy with the amendments made, and believe the authors have addressed the issues.

Comment by Reviewer 1:

“I did not find the comparison with the instability theorem of [50] was quite correct. The authors state that the inequality of [50] does not hold for all time, so they should not use it. However, that was the point of my suggestion - that the stability/instability should hold for some times, not all times (e.g., over some time interval I_k , where each I_k will be different for the k -th Turing mode). This tells one when a Turing instability will be present, leading to patterns. When the instability is suppressed, the inequality changes sign, and then the solutions instead tend to a spatially homogeneous state.

In the present paper, the authors find that their system evolves between a patterned state and a spatially homogeneous state (as shown in some figures and movies). This means that each state must gain and lose stability over different time intervals. So, no Turing mode should be unstable for all time under such conditions. In particular, perturbations of the $k=0$ state against Turing modes for $k \geq 1$ should give instabilities when the dynamics tend to the patterned state, while when the dynamics are spatially uniform all these perturbations should be stable. This means that the inequality in [50] corresponding to such a system will change sign.

Now, if we compare this to the semi-analytic approach of the authors, we see that the authors always find an instability result which is independent of time. This would suggest a pattern which emerges and then persists for all time, yet this is not what the numerical simulations show. Let us look, for example, at Figure 3a(ii-iv). In this figure, we see that the oscillatory motion suppresses the Turing instability (panel iii), and after it is switched off the Turing instability appears again (panel iv). The instability analysis of the authors would not predict this absence of patterns for the region where oscillation are switched on, since it neglects time.

In their response, the authors state:

"The condition shown in Eq. (12) in the paper by van Gorder is a sufficient condition for the instability for the perturbation with wavenumber k . We have numerically evaluated the inequality in Eq. (12) and found that it does not hold for a certain time within a period at the wavenumber $k \approx 1.0$ for both $A = 1$ and 2 . It should be noted that the uniform state is stable and unstable for $A = 1$ and 2 , respectively. Considering that, the theorem claims that if the inequality holds for all time then the system becomes unstable for the

perturbation with a wavenumber k . Therefore, the theorem in the paper by van Gorder cannot determine the boundary of the instability in our system."

They also state:

"In both the cases inequality (5.12) does not hold for certain ranges in t , which means that inequality (5.12) cannot conclude whether the system is stable or not."

The mistake of the authors is they assume that the boundary of the stability region is fixed over time. In contrast, the stability region itself must vary in time, in order to generate the alternation between the presence and absence of patterns as seen in fig. 3(a) and some of the movies.

Indeed, the results discussed in [50] only hold for certain ranges of time, rather than all time, by design. This allows one to consider the subset of times for which a given Turing mode is unstable, and hence to understand patterns which are only present some and not all of the time, just like the patterns in the paper under review. Looking at the examples in [50], we see that a change in the instability region tends to correspond to a change in the pattern present over time. This was also true in the paper [35], where a change in boundary temperature over time resulted in a change in the instability region and hence of the Turing patterns over time. See Figures 11, 12, and 13 in that paper - all of which feature time-varying instability regions corresponding to Turing patterns which change in time.

Part of the reason the authors may be confused is that they are seeking temporal eigenvalues in order to determine stability/instability of the Turing modes. However, it is well known that eigenvalues are no longer useful indicators of stability for systems with time-varying parameters; see, for instance,

<http://www.chebfun.org/examples/ode-linear/FrozenCoeffs.html>

O. Perron, Die Stabilitätsfrage bei Differentialgleichungen, Math. Zeit. 32 (1930), 703-728.

M. Wu, A note on stability of linear time-varying systems, IEEE Trans. Autom. Control 19 (1974), 162-162.

Josić K, Rosenbaum R. 2008 Unstable solutions of nonautonomous linear differential equations. SIAM Rev. 50, 570-584.

I would suggest the authors think a bit more about these points regarding the stability

analysis. As it stands, the results of their stability analysis stand in contrast to the results of their numerical simulations.”

Author reply: The authors would again like to thank the reviewer for their reading of our manuscript. It is possible that there is a misunderstanding regarding our analysis. The reviewer has considered that the authors only discuss the stability of the steady state based on the eigenvalues of the time-change linearized operator for each wave number k . Here, since our system has a periodic external force, normal analysis of the Turing instability of course gives us the conclusion that the system is sometimes stable, and sometimes unstable. We understand that the stability/instability condition for each time does not give us any information about the stability of the system with a periodic forcing.

In contrast, we have considered the (linearized) Poincare map $\mathbf{P}(k)$, which gives us the time evolution of the system over one period of the external forcing. Since it is difficult to analytically calculate the Poincare map, we took a semi-analytical approach. In particular, we numerically obtained the Poincare map, and discussed the stability of the system under the periodic forcing; i.e., the stability of the periodic uniform solution. The newly obtained figure was added as Figure 5.

Relatedly, we have noticed that typo has arisen during the editing process. Equation 28 in the Appendix should be written in matrix form (a new_line before (σb) was deleted, and $\mathbf{L}(t;k)$ does not appear as a matrix). We used bold the symbol to indicate matrixes in the text. We have also missed the symbol for the transposed matrix " t " in the equation above Figure 4 and Equations 29 and 30. The authors apologize for any misunderstanding this may have caused. The revision fixes this issue. We also added the description on the mathematical meaning of our analysis in the Section "Linear Stability analysis" as:

In our investigation, we perform the linear stability analysis on the uniform oscillatory solution for the system in Equations (1) and (2). $\mathbf{P}(k)$ is a linearized Poincare map for a period T , which shows the time evolution of the perturbation with the wave number k imposed to a uniform oscillatory solution.

and in the Appendix (just below Equation 30) as:

Equation (30) is considered as an autonomous time-discrete dynamical system for the perturbation added to the uniform oscillatory state with the spatial wave number k . In other words, $\mathbf{P}(k)$ is a linearized Poincare map for the perturbation with the wave number k .

We also added reference [51] to the comment of the reviewer to clarify this issue.

Comment by Reviewer 2:

“The authors have responded to my comments well. They have conducted additional simulations and stability analysis to clarify the bistability and oscillation frequency dependence. I recommend this manuscript for publication if the following minor points are addressed.”

Author reply: We thank the Reviewer for his/her comments and the positive assessment of our work.

“The coexistence of two types of Turing patterns is shown in Fig. S6, whereas a homogeneous oscillation phase is shown above $a = 27$ in Fig. S5. Is this difference due to hysteresis? Do the Turing and oscillation phases coexist at $27 < a < 40$? It should be specified. The explanation in the text is not sufficient. In Page 7, the sixth line from Eq.(2), “We found that the Turing pattern formation occurs between $a=23$ and $a=27$ from spots to stripes. Below $a=23$ and above $a=27$, stationary homogeneous patterns and homogeneous oscillations exist, respectively”. This is only for a specific initial condition.”

Author reply: We thank this important comment. The difference between the figures is the initial condition. Figure S5 shows the results starting from the same homogeneous initial state. However, Figure S6 represents the observations made by starting each step from the result of the previous one. Definitely, once the patterns are formed they persist over a wide range of parameters, and we found multistability. We corrected the text on page 7 and the figure captions.

“A larger spacing grid and longer Delta t are used for the simulations for new figures. If they do not change the results, it is OK but should be mentioned somewhere.”

Author reply: To save the simulation time, in the new simulations, we used greater grid spacing and time step. However, it does not change the obtained results. To indicate this information, we rewrote the corresponding part in the “Numerical simulations” part.

“It is written that the initial state of Fig. S7(b) is a Turing state. It is probably a spot pattern. The authors should specify it more exactly.”

Author reply: The Reviewer is right. The Turing state is in which any patterns appear (spotted, striped or their combination). We added this information to the figure caption (Figure S7).

“The section of the Numerical model in Page 15 should be updated. Different spacing, time steps, and initial conditions are also used in the additional simulations.”

Author reply: This is a valid point. We rewrote the corresponding part in this section.

Comment by Reviewer 3:

“I am happy with the amendments made, and believe the authors have addressed the issues.”

Author reply: We thank the Reviewer for his/her comments and the positive assessment of our work.

This completes our reply. We again thank the Referees for their comments/suggestions and critical reading of our work. We hope that the manuscript is now in an acceptable form in *Communications Chemistry*.

REVIEWERS' COMMENTS:

Reviewer #1 (Remarks to the Author):

The authors consider a periodically forced chemically-motivated system which exhibits some interesting behaviors as it evolves in time. I feel the work is valuable, and will perhaps motivate future experimental and theory work.

While I'm still not sure the approach to the stability analysis is the most rigorous approach, I understand that this is a chemistry journal and not a mathematics journal, so I will not belabour this point any further. At the very least, the authors make a compelling case through their experiments and simulations.

I feel the work may be accepted in its present form.

Reviewer #2 (Remarks to the Author):

The authors have revised the manuscript well. I recommend it for publication.

Reviewer #1 (Remarks to the Author):

The authors consider a periodically forced chemically-motivated system which exhibits some interesting behaviors as it evolves in time. I feel the work is valuable, and will perhaps motivate future experimental and theory work.

While I'm still not sure the approach to the stability analysis is the most rigorous approach, I understand that this is a chemistry journal and not a mathematics journal, so I will not belabour this point any further. At the very least, the authors make a compelling case through their experiments and simulations.

I feel the work may be accepted in its present form.

Authors reply: We thank for the positive evaluation of our work.

Reviewer #2 (Remarks to the Author):

The authors have revised the manuscript well. I recommend it for publication.

Authors reply: We thank for the positive evaluation of our work.